# Osteochondral Tissue-On-a-Chip: A Novel Model for Osteoarthritis Research

**DOI:** 10.3390/ijms25189834

**Published:** 2024-09-11

**Authors:** Irene González-Guede, Daniel Garriguez-Perez, Benjamin Fernandez-Gutierrez

**Affiliations:** 1UGC de Reumatología, Hospital Clínico San Carlos, IdISSC, 28040 Madrid, Spain; igguede@salud.madrid.org; 2UGC de Traumatología y Cirugía Ortopédica, Hospital Clínico San Carlos, IdISSC, 28040 Madrid, Spain; daniel.garriguez@salud.madrid.org; 3Facultad de Medicina, Universidad Complutense de Madrid, 28040 Madrid, Spain

**Keywords:** Tissue-On-a-Chip, Organ-On-a-Chip, Joint-On-a-Chip, osteochondral tissue, osteoarthritis, cartilage, bone

## Abstract

The existing in vitro and in vivo models for studying osteoarthritis have significant limitations in replicating the complexity of joint tissues. This research aims to validate a Tissue-On-a-Chip system for osteoarthritis research. Osteochondral tissues obtained from knee replacement surgeries of patients with osteoarthritis were cultured in an Organ-On-a-Chip system. This system was designed to supply oxygen and glucose to the cartilage from the bone. The distribution of oxygen and glucose was evaluated by fluorescence using Image-iT Green Hypoxia and 2-NBDG, respectively. Cytotoxicity was measured using lactate dehydrogenase (LDH) levels in chip cultures compared to plate cultures (12 tissues per method). Glycosaminoglycans (GAGs), alkaline phosphatase (ALP), Coll2-1, and procollagen type II N-terminal propeptide (PIINP) were measured in the perfused medium of the Tissue-On-a-Chip over a period of 70 days. Fluorescence of Image-iT Green Hypoxia was observed only in the cartilage area, while 2-NBDG was distributed throughout the tissue. An increase in LDH levels was noted in the plate cultures on day 24 and in the Tissue-On-a-Chip cultures on day 63. Compared to the start of the culture, GAG content increased on day 52, while ALP showed variations. A notable increase in GAG, ALP, and Coll2-1 levels was observed on day 59. PIINP levels remained stable throughout the experiment. The validated osteochondral Tissue-On-a-Chip system can replicate the joint microenvironment, with hypoxic conditions in cartilage and normoxic conditions in bone. Tissue survival and component stability were maintained for approximately two months. This platform is a useful tool for evaluating new drugs and represents a viable alternative to animal models.

## 1. Introduction

Osteoarthritis (OA) is characterized by the loss of articular cartilage and the progressive thickening of the subchondral bone [1]. Globally, osteoarthritis of the knee and hip is a leading cause of disability, with a dynamically increasing prevalence due to an aging population and rising risk factors [2]. Despite being a major public health concern, there is currently no effective treatment for OA. Available treatments primarily focus on alleviating pain and stiffness; in severe cases, total joint arthroplasty is performed [3].

To address this issue, research is focused on developing therapeutic strategies to reduce or reverse the effects of osteoarthritis. However, current in vitro and in vivo models have significant limitations. In terms of cell cultures, they do not accurately reflect the complexity of joint tissues. Chondrocytes, the cells found in cartilage, require a 3D culture environment to reproduce the extracellular matrix (ECM) and maintain hypoxic conditions. The ECM provides volume to the cartilage and consists of water, collagen, proteoglycans, non-collagenous proteins, and glycoproteins [4,5]. Additionally, because cartilage is an avascular and aneural tissue, nutrients and oxygen are supplied from the subchondral bone and synovial fluid. Therefore, co-cultures of cartilage with subchondral bone or synovial membrane are necessary [6,7]. Currently, three-dimensional tissue cultures or organoids are proposed to facilitate communication between tissues and better mimic morphology and physiological functions, thus yielding more reliable results. However, these co-cultures often fail to maintain cell viability over extended periods and do not replicate the varying oxygen conditions present in vivo, leading to phenotypic changes in chondrocytes when exposed to normoxic conditions [8].

In in vivo studies, there is no single standard animal model that faithfully replicates all aspects of osteoarthritis in humans. Although small animals are commonly used, the results must ultimately be validated in large animals due to their anatomical similarities to humans. Large animals, however, present disadvantages related to management, cost, slower disease progression, and ethical considerations [9]. Additionally, no animal model perfectly captures all features of OA, as they only exhibit some characteristics of the disease, such as pain, synovitis, cartilage degeneration, and bone remodeling [10].

Given the limitations of current models for studying osteoarthritis, there is a need for a more reliable, accessible, and reproducible model to advance research in this field. In recent years, an advanced 3D culture technology known as Organ-On-a-Chip (OOC) has been developed. These microfluidic devices for 3D cell culture feature continuous perfusion chambers, ensuring an optimal supply of nutrients and preventing the accumulation of waste products [11]. This novel technology is highly versatile and can recreate complex physiological environments, making it a promising alternative for basic research and drug studies, potentially replacing animal models [12].

The objective of this research is to validate a microfluidic system of osteochondral tissue derived from knee osteoarthritis patients undergoing joint replacement, using Organ-On-a-Chip technology for the study of osteoarthritis. The proposed design and assembly of this device aim to recreate the microenvironment of cartilage and bone tissues, maintaining communication between tissues and cell viability over a prolonged period.

## 2. Results

### 2.1. Oxygen and Glucose Supply in Osteochondral Tissue-On-a-Chip

Osteochondral tissues were cultured in the designed osteochondral Tissue-On-a-Chip device at a flow rate of 2 µL/min. Due to the device’s gas isolation properties, oxygen was supplied exclusively through the culture medium. In our Organ-On-a-Chip model, the osteochondral tissue is positioned such that the bone lies closer to the membrane in contact with the culture medium, while the cartilage is positioned above it. This arrangement potentially creates an oxygen gradient, with lower oxygen levels in the cartilage, mimicking physiological conditions. To verify that the cartilage in our model was indeed under hypoxic conditions, we used the fluorescent reagent Image-iT Green Hypoxia, which specifically marks hypoxic regions. As shown in Figure 1A,B, only the cartilage layer exhibited fluorescence. Ensuring that the osteochondral tissue, particularly the cartilage, received an adequate glucose supply despite the oxygen limitation was crucial. For this purpose, we used 2-NBDG, a fluorescent glucose marker. Figure 1C,D demonstrate that both the cartilage and bone exhibited fluorescence, indicating glucose uptake in these tissues.

### 2.2. Viability and Stability of Osteochondral Tissue

#### 2.2.1. Osteochondral Tissues

To assess cellular cytotoxicity and biomarkers in the tissues, knee samples from three patients undergoing surgery were used: a 64-year-old female, a 63-year-old male, and a 79-year-old female. Samples were taken from the injured regions with grade 2 or 3 lesions, as well as from the uninjured areas of the femur and tibia of each patient (n = 12 osteochondral tissues).

#### 2.2.2. Lactate Dehydrogenase

A lactate dehydrogenase (LDH) assay was performed to assess cellular cytotoxicity. For this purpose, 12 osteochondral tissues cultured on a plate were compared with 12 tissues cultured in the Tissue-On-a-Chip model. LDH levels were determined in a culture medium collected every 3 or 4 days, comparing each culture method with the initial levels measured at 3 days. In plate cultures, a significant increase in LDH was observed on days 24 and 28. In the Tissue-On-a-Chip model, an increase in LDH began to be evident on days 63, 66, and 70, showing a gradual rise (Figure 2).

#### 2.2.3. Biomarkers in Osteochondral Tissue-On-a-Chip

Levels of various biomolecules related to tissue stability were evaluated from the perfused medium collected over 70 days of culture in the osteochondral Tissue-On-a-Chip. Temporal variations were analyzed relative to the start of the culture on day three. The graphs in Figure 3 depict the evolution of the concentrations of the studied biomolecules throughout the culture period. Glycosaminoglycan (GAG) content increased at 52 days and continued to rise on day 59 until the end of the culture. Alkaline phosphatase levels showed a decrease on days 14 and 17, followed by an increase on day 31 and a marked increase on day 59. Coll2-1 content also rose starting on day 59, while PIINP levels remained stable throughout the culture period.

## 3. Discussion

Organ-On-a-Chip (OOC) technology represents a significant advancement in the study of various diseases. These devices replicate the microarchitecture and function of human organs in a controlled environment, providing a more physiologically relevant model than traditional methods [13]. The use of OOCs, and tissue chips in particular, is an advanced platform for drug discovery and toxicity assessment [14], with the potential to reduce reliance on animal testing, in line with the 3Rs principles (replace, reduce, refine) [15].

Joint-On-a-Chip (JOC) systems are specifically designed to study joints. These devices typically consist of tissues formed from different cell types, and some are cultured in 3D with or without scaffolds [16,17,18]. While these JOCs are a breakthrough for research, they often lack true communication between joint tissues such as cartilage, subchondral bone, and the synovial membrane.

In our Organ-On-a-Chip model, we utilized osteochondral tissue from osteoarthritis patients, enhancing the relevance of the model for translational research. The close relationship between subchondral bone and cartilage is crucial for the biochemical and molecular interactions essential for osteoarthritis homeostasis and progression. Evidence suggests direct interactions through vascular channels and microfissures that allow communication between chondrocytes and subchondral bone cells [19]. The co-culture of cartilage and subchondral bone in our OOC maintains natural interactions between these tissues and better replicates physiological conditions. Incorporating patient-specific factors such as age, gender, and genetic conditions adds precision to the studies. Additionally, including pathological conditions provides a more realistic model of osteoarthritis. Traditional in vitro and in vivo models often involve chemically or surgically induced disease, which does not fully replicate the condition [8,9].

Cartilage, an avascular tissue, relies on subchondral bone for nutrients and oxygen. This relationship is critical for maintaining cartilage viability and function [6,7]. Hypoxia is essential for chondrocyte survival and function, with oxygen tensions ranging from 1% to 5%. In contrast, subchondral bone cells exist under normoxic conditions in the physiological environment [20]. In our validated Tissue-On-a-Chip model, the subchondral bone was placed on the membrane to provide nutrients to the cartilage. The chip’s gas-impermeable materials ensure that oxygen is delivered from the culture medium through the bone to the cartilage. Given the considerable height of the osteochondral tissue, this configuration helps regulate oxygen levels for the cartilage. In vivo fluorescence assays confirmed that the cartilage remains hypoxic, mirroring physiological conditions. Additionally, the fluorescent glucose marker was effectively distributed throughout the osteochondral tissue, fulfilling its metabolic needs even under hypoxic conditions.

We assessed tissue survival by evaluating cell cytotoxicity. Hypoxia of less than 1% in chondrocytes impacts their basal metabolic activity. Proper glucose control is also crucial; excess glucose can lead to cell death, while a deficiency can inhibit oxygen consumption [20,21]. LDH activity was stable for up to 63 days in our osteochondral Tissue-On-a-Chip compared to 24 days in plate culture, indicating that our chip design and parameters support prolonged tissue cell survival.

We monitored ECM-related markers in the chip’s perfused medium for 70 days. Glycosaminoglycans (GAGs), essential components of the cartilage ECM, bind to proteins to form proteoglycans that maintain cartilage structure and function. Collagen type II, another key ECM component, forms a network providing structural support. In the chip-cultured osteochondral explants, GAG levels decreased at 52 days but stabilized and increased on day 59 onward. Coll2-1 levels, indicating type II collagen degradation, rose on day 59. The observed degradation of GAGs and Coll2-1 suggests a loss of cartilage structure during this period. The stability of PIINP, a marker of type II collagen synthesis, implies that the tissue continues to attempt collagen synthesis despite the observed degradation.

Alkaline phosphatase (ALP), an enzyme produced by osteoblasts, is an indicator of bone mineralization and structure [22]. In chip culture, ALP levels decreased at two weeks but increased at one month, suggesting ongoing bone remodeling. A significant increase in ALP at day 59 coincides with changes in other biomarkers, potentially indicating abnormal bone remodeling.

Overall, our study indicates that ECM degradation occurs in cartilage and subchondral bone around day 59 of chip culture, likely contributing to the cellular cytotoxicity observed at 63 days. This ECM degradation negatively impacts the cellular environment, impairing chondrocyte and bone cell viability and function.

Our study has limitations. Oxygen and glucose distribution throughout the tissue have not been quantified. Additionally, this model lacks biomechanical components, and defining the appropriate pressure for osteochondral tissue remains challenging. Excess or insufficient pressure could affect tissue survival.

In conclusion, the osteochondral Tissue-On-a-Chip offers a valuable tool for studying osteoarthritis, remaining stable for approximately two months. The co-culture of cartilage and subchondral bone provides a more realistic disease model, incorporating patient-specific factors. This is the first osteochondral explant model in an Organ-On-a-Chip and the first to naturally recreate a hypoxia–normoxia environment in cartilage–bone tissue without nitrogen application. The adequate supply of nutrients and oxygen, the importance of hypoxia for chondrocytes, and the stability of key biomarkers suggest that this model could enhance osteoarthritis research, align with the 3Rs principles, and advance pharmacological research and personalized medicine.

## 4. Materials and Methods

### 4.1. Patients

Osteochondral tissue samples were provided by the Traumatology and Orthopedic Surgery Unit at Hospital Clínico San Carlos, Madrid, Spain. The tissues were obtained from patients diagnosed with osteoarthritis according to the 1992 American College of Rheumatology (ACR) criteria and who had undergone joint replacement surgery. Prior to surgery, patients were informed about this study and provided written informed consent. The study protocol was approved by the Institutional Ethics Committee (Comité Ético de Investigación Clínica, Hospital Clínico San Carlos, Madrid, Spain) in accordance with the principles of the Declaration of Helsinki under study code 21/133-E.

### 4.2. Osteochondral Tissue-On-a-Chip System

Figure 4A shows a schematic of the overall setup of the osteochondral Tissue-On-a-Chip. The materials used for the Organ-On-a-Chip system include cyclic olefin polymer (COP) and cyclic olefin copolymer (COC) (Be-Transflow, BeOnChip). These medical-grade plastics exhibit very low oxygen permeability. The inlet and outlet tubes are made of PET (BeOnChip), which also features low gas permeability, and are connected to the chip via connectors (Figure 4B). A syringe with a 20G needle is inserted into the inlet tube and placed in a perfusion pump (New Era Pump Systems, Inc., 138 Toledo Street, Farmingdale, NY, USA) set to a flow rate of 2 µL/min. The perfused medium is collected through the outlet tube for subsequent analysis (Figure 4C).

A porous membrane with an 8-micron pore size is positioned between the culture medium and the sample. The osteochondral tissue is placed with the bone side against the membrane. To ensure a complete seal of the circuit, an adhesive lid is applied.

In this gas-impermeable setup, the oxygen supplied to the sample comes from the oxygen dissolved in the culture medium, which is controlled by the selected flow rate. Additionally, the subchondral bone distributes nutrients and oxygen to the cartilage, mimicking physiological conditions.

Osteochondral tissues were prepared for culture by cutting them into dimensions of 3 mm in length and width and 4 mm in height. An acrylic cutting block with 1 mm sections was used as a guide, and carbon steel blades were employed for cutting (Figure 4D). The explants were hydrated with PBS and 1% Pen/Strep during the cutting process.

### 4.3. Fluorogenic Hypoxia Detection

The Image-iT Green Hypoxia reagent (Thermo Fisher, Waltham, MA, USA) was used to detect hypoxia in live cells. This reagent is an irreversible fluorogenic marker indicating oxygen concentrations below 5%. After 24 h of cultivation, a syringe with 0.5 µL of the reagent from a 5 µM stock solution in 500 µL of culture medium was attached to the chip. The flow rate was adjusted to 2 µL/min and maintained for three hours. After this incubation period, tissues were removed from the chip and fixed in 4% formaldehyde for 20 min. The tissues were then washed with PBS and incubated with DAPI (1:1000, Thermo Fisher) for 10 min. Following additional washes with PBS, the tissues were cut into 1 mm thick sections and visualized with a confocal microscope (Fluoview 1000, Olympus, Tokyo, Japan).

### 4.4. Glucose Distribution

To evaluate glucose distribution in the tissue, 2-NBDG (2-(N-(7-nitrobenz-2-oxa-1,3-diazol-4-yl) amino)-2-deoxyglucose) (Thermo Fisher) was used. 2-NBDG is a fluorescent glucose analog used to monitor glucose uptake in live cells. The tissues cultured for 24 h were removed from the chip and incubated in the reagent at 300 µM for one hour. The tissues were then washed with HBSS (1X) and cut into 1 mm sections. Images were taken using a confocal microscope (Fluoview 1000, Olympus).

### 4.5. Osteochondral Tissue Culture

Osteochondral tissues from three patients were cultured using the Organ-On-a-Chip system and standard plates. The femur and tibia from both the lesion and non-lesion areas were cultured (n = 12 osteochondral tissues per culture method). The culture medium used was DMEM with GlutaMAX (Gibco-Thermo Fisher, Waltham, MA, USA), 10% FBS (Gibco), and 1% Pen/Strep (Gibco). The culture medium was collected every three or four days. Both methods were maintained in an incubator at 37 °C and 5% CO_2_.

### 4.6. Lactate Dehydrogenase Assay

The cytotoxicity of explants cultured on plates and Organ-On-a-Chip was evaluated using the CyQUANT LDH Cytotoxicity Assay Kit (Thermo Fisher) according to the manufacturer’s instructions. For comparison of both culture methods, 50 µL of perfused medium collected from the chip and 11.11 µL and 8.33 µL of supernatant from the plate culture corresponding to three and four days of culture, respectively, were analyzed. Absorbance was measured with a FLUOstar Omega (BMG Labtech, Ortenberg, Germany). Prior to analysis, absorbance values at 490 nm and 690 nm were subtracted.

### 4.7. Biochemical Determinations of Tissue-On-a-Chip Perfused Medium

Glycosaminoglycan (GAG), alkaline phosphatase (ALP), Coll2-1, and procollagen type II N-terminal propeptide (PIINP) levels were determined in the culture medium perfused through the Tissue-On-a-Chip. Glycosaminoglycan content was measured using the Human GAGs (Glycosaminoglycan) ELISA Kit (Ref. E-EL-H2598, MyBiosource, San Diego, CA, USA). Alkaline phosphatase was determined using PNPP—phosphatase substrate (Ref. 37620, Thermo Fisher). Cartilage degradation and synthesis were assessed using the Human Coll2-1 ELISA Kit (Ref. CSB-EQ027311HU, Cusabio, Houston, TX, USA) and the Human Procollagen II N-Terminal Propeptide (PIINP) ELISA Kit (Ref. BSKH61221, Bioss Antibodies, Woburn, MA, USA), respectively. All kits were used according to the manufacturer’s instructions, and absorbance was measured with a FLUOstar Omega (BMG Labtech, Ortenberg, Germany).

### 4.8. Statistical Analysis

Statistical analysis was performed using GraphPad Prism 8.0 software. Outliers were detected using the ROUT method (Q = 1%). The D’Agostino and Pearson normality test was used to determine if the data followed a normal distribution. If the data followed a normal distribution, values over time for each culture method were compared with the initial time (3 days) using a one-way ANOVA test, with a significance level of 0.05. If the data did not follow a normal distribution, the Kruskal–Wallis test was used.

## 5. Patents

Specifically, a European patent has been filed titled “OSTEOCHONDRAL TISSUE ON A CHIP SYSTEM AND METHOD FOR CULTURING OSTEOCHONDRAL TISSUE” with application number EP24382757, covering aspects of the technology and methodologies described herein. I.G.-G. and B.F.-G. are listed as the inventors of the patent.

## Figures and Tables

**Figure 1 ijms-25-09834-f001:**
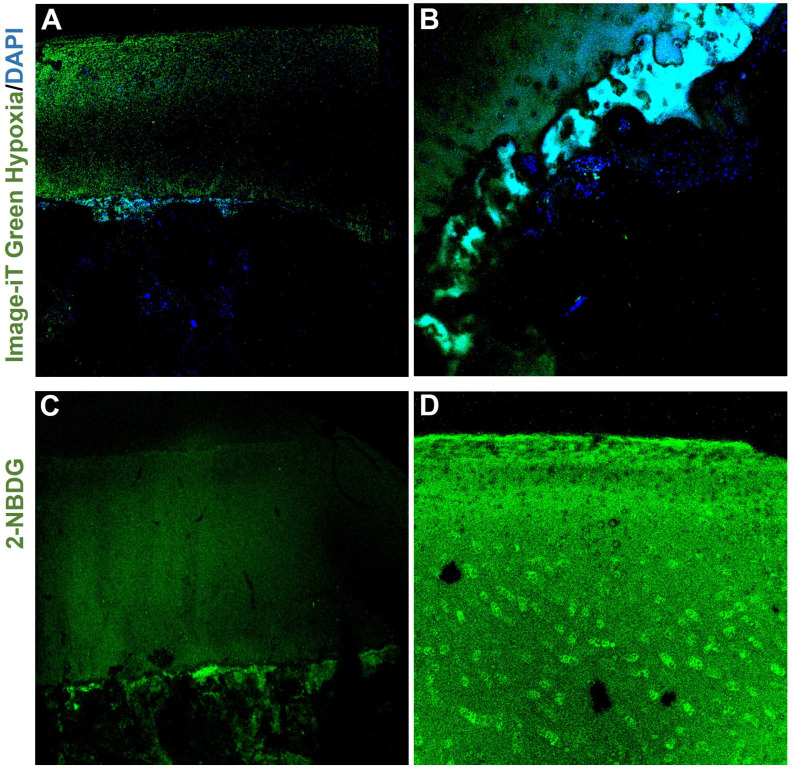
Osteochondral tissues cultured in a Tissue-On-a-Chip device. Hypoxic regions were labeled with Image-iT Green Hypoxia, as shown in (**A**,**B**), where fluorescence was observed in the cartilage layer. Bone cells were stained only with DAPI (images captured with 4× and 10× objectives, respectively). Glucose supply was assessed using 2-NBDG; fluorescence was observed in both cartilage and bone, as shown in (**C**). In (**D**), a detailed view of the cartilage is presented (images captured with 4× and 10× objectives, respectively).

**Figure 2 ijms-25-09834-f002:**
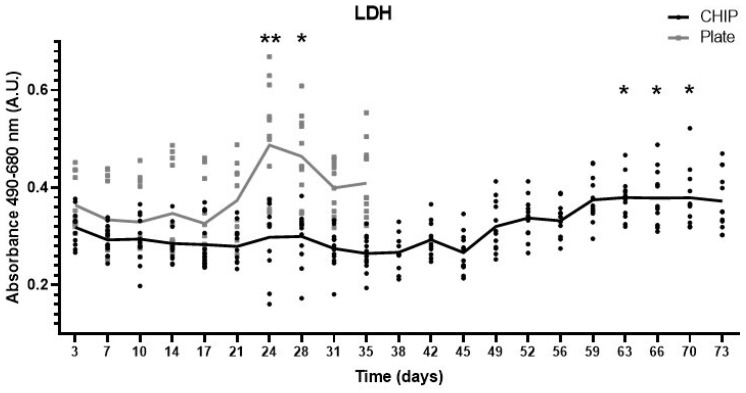
LDH levels in osteochondral tissue cultured on plates and in Tissue-On-a-Chip. Femur and tibia explants, both lesioned and non-lesioned, from the knees of three patients were used (n = 12 per culture method). A one-way ANOVA test was used to compare each time point with the baseline at day 3: * *p* < 0.05, ** *p* < 0.01.

**Figure 3 ijms-25-09834-f003:**
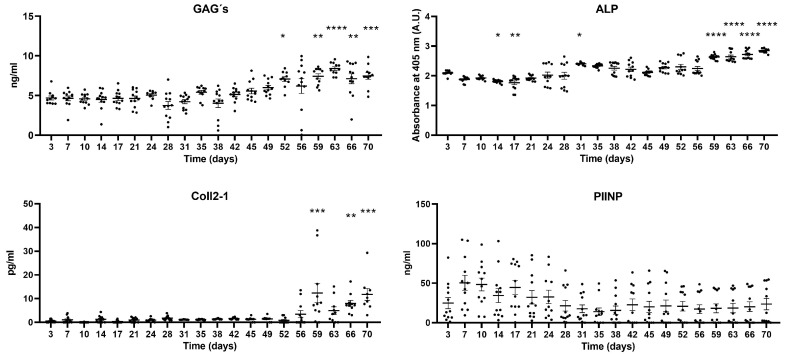
GAG, ALP, Coll2-1, and PIINP content in the perfused medium of the osteochondral Tissue-On-a-Chip determined by ELISA over 70 days. Femur and tibia explants, both lesioned and non-lesioned, from the knees of three patients were used (n = 12). A one-way ANOVA test was used to compare each time point with the baseline at day 3: * *p* < 0.05, ** *p* < 0.01, *** *p* < 0.001, **** *p* < 0.0001. All bar graphs are presented as mean ± SEM.

**Figure 4 ijms-25-09834-f004:**
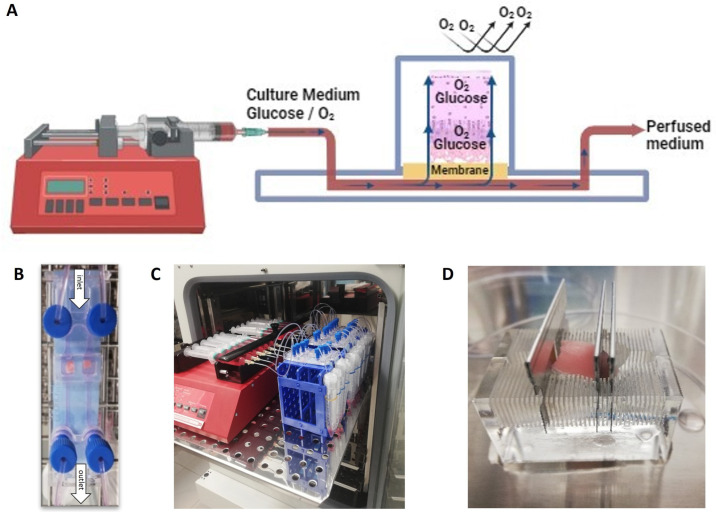
Osteochondral Tissue-On-a-Chip Setup. (**A**) Diagram of the osteochondral Tissue-On-a-Chip. The Organ-On-a-Chip materials used are gas tight, so oxygen is supplied solely from the culture medium. A syringe perfuses the culture medium at a flow rate of 2 µL/min. The subchondral bone of the osteochondral tissue is placed on the membrane, distributing glucose and oxygen to the cartilage, mimicking physiological conditions. Created with BioRender.com. (**B**) Organ-On-a-Chip model. The chip is composed of cyclic olefin polymer (COP) and cyclic olefin copolymer (COC). PET inlet and outlet tubes, connected with connectors, facilitate perfusion. (**C**) Assembly of the osteochondral Tissue-On-a-Chip. The photograph shows multiple chips connected to syringes inside an incubator, with tubes collecting the culture medium. (**D**) Cutting of osteochondral explants. The explants are sectioned into 3 × 3 × 4 mm pieces using an acrylic cutting block and carbon steel blades.

## Data Availability

The data presented in this study are available upon request from the corresponding author.

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
