# Peer review of "Osteochondral Tissue-On-a-Chip: A Novel Model for Osteoarthritis Research"

_ijms, 2024, doi:10.3390/ijms25189834_

Round 1

Reviewer 1 Report

Comments and Suggestions for Authors

ijms-3175506

Osteochondral Tissue-On-a-Chip: A Novel Model for Osteoarthritis Research

This article describes the development of a new, reliable organ-on–a-chip system for studying osteoarthritis (OA) using human osteochondral tissue samples. The system was designed to adequately reflect the distribution of oxygen and nutrients under physiological conditions (e.g., to maintain hypoxia in the cartilage). The data presented in this study show that the model provides stable conditions up to two months. Thus, the authors conclude that the osteochondral tissue-on-a-chip represents a promising tool for the analysis of OA.

The newly developed tissue-on-a-chip model is well designed. The study is carried out properly and technically sound. The manuscript is excellently written and the presentation of the data is straightforward and clear. The results are comprehensible and conclusive. Thus, there are only a few points/questions requiring the authors’ consideration.

1.            According to the information in the Materials and Methods, oxygen and glucose supply in the tissue were assessed after 3 h in the tissue-on-a-chip chamber. Did the authors check whether the distribution of oxygen and glucose is stable over a longer time (i.e., up to the 60 days)?

2.            The figures should be numbered in the order of their appearance in the text.

3.            Section 2.2.1: Please comment on the amount of tissue injury in the injured tissue samples used.

4.            Section 2.2.2/Figure 3: Why are the data derived from the plate model only shown up to day 35? Based on the text, I would assume that tissue samples in both models were cultivated for 73 days.

5.            Sections 2.2.2 and 2.2.3: Was there a difference in lesioned and non-lesioned samples, respectively (either in the plate or in the chip)? Or among samples from women and the man?

6.            Figures 3 and 4: In respect of the - in part relatively high - variability at each time point, I see the necessity of standardizing the quality of the tissue samples used (otherwise, the results obtained with this model would be difficult to compare between different labs). Is there an approach for such standardisation?

7.            Section 4.2: Did the oxygen-impermeable chamber, in which the tissue samples were cultivated, contain some remaining air/oxygen in the beginning (i.e., was there some oxygen that may disturb hypoxic conditions at the start of the experiment)? Or did the authors remove gas/oxygen from the chamber?

Author Response

Reviewer 1

This article describes the development of a new, reliable organ-on–a-chip system for studying osteoarthritis (OA) using human osteochondral tissue samples. The system was designed to adequately reflect the distribution of oxygen and nutrients under physiological conditions (e.g., to maintain hypoxia in the cartilage). The data presented in this study show that the model provides stable conditions up to two months. Thus, the authors conclude that the osteochondral tissue-on-a-chip represents a promising tool for the analysis of OA.

The newly developed tissue-on-a-chip model is well designed. The study is carried out properly and technically sound. The manuscript is excellently written and the presentation of the data is straightforward and clear. The results are comprehensible and conclusive. Thus, there are only a few points/questions requiring the authors’ consideration.

Comments 1: According to the information in the Materials and Methods, oxygen and glucose supply in the tissue were assessed after 3 h in the tissue-on-a-chip chamber. Did the authors check whether the distribution of oxygen and glucose is stable over a longer time (i.e., up to the 60 days)?

Response 1: We would like to thank the reviewer for their positive feedback and for all the useful comments and suggestions. In response to comment 1, tissues were cultured for 24 hours on the organ-on-a-chip, and subsequently stained. We introduce this clarification in section 4.3. and in section 4.4:

“After 24 hours of cultivation, a syringe with 0.5 µL of the reagent from a 5 µM stock solution in 500 µL of culture medium was attached to the chip” (line 258-259).

“The tissues cultured for 24 hours were removed from the chip and incubated in the reagent at 300 µM for one hour”(line 268-269). “

Since the device is impermeable to gases and provides a constant supply of culture medium, we consider the distribution of oxygen and glucose to be stable throughout the culture period.

Comments 2: The figures should be numbered in the order of their appearance in the text.

Response 2: Thank you for noting this. We have renumbered the figures to ensure they follow the correct order of appearance in the text and figure legends.

Comments 3: Section 2.2.1: Please comment on the amount of tissue injury in the injured tissue samples used.

Response 3: We have now specified the grade of lesions in the section 2.2.1.: “Samples were taken from the injured regions with grade 2 or 3 lesions, as well as from the uninjured areas of the femur and tibia of each patient” (Line 104-105).

Comments 4:   Section 2.2.2/Figure 3: Why are the data derived from the plate model only shown up to day 35? Based on the text, I would assume that tissue samples in both models were cultivated for 73 days.

Response 4: Thank you for this observation. We have removed the phrase "for 73 days" from Section 2.2.2 (line 109) and the figure legend. Data are shown until the tissue samples showed significant deterioration on plate and chip.

Comments 5:  Sections 2.2.2 and 2.2.3: Was there a difference in lesioned and non-lesioned samples, respectively (either in the plate or in the chip)? Or among samples from women and the man?

Response 5: Thank you for this question. When analyzing by groups the tissues with lesion and without lesion with respect to culture start, the significance is similar at the same days as when analyzing both groups together. On the other hand, comparing each group over time could be interesting; however, since this was not the aim of our research, we did not select all lesion grades (0-4) for each patient, and the number of females and males is not sufficient for a comparative analysis. Therefore, the results may not be conclusive.

Comments 6: Figures 3 and 4: In respect of the - in part relatively high - variability at each time point, I see the necessity of standardizing the quality of the tissue samples used (otherwise, the results obtained with this model would be difficult to compare between different labs). Is there an approach for such standardisation?

Response 6: We agree with the reviewer that standardizing the quality of tissue samples is crucial to ensure reproducibility in different laboratories. The procedure we followed is detailed in the manuscript. Variability in LDH levels is mainly observed in plate cultures. In the osteochondral tissue-on-a-chip model, there is less variability and the increase in cytotoxicity is more gradual. Biomarkers ALP, GAGs and Coll2-1 in osteochondral tissue-on-a-chip culture show a marked increase at the start of degradation in all tissues (59 days). We consider our model to be replicable in other laboratories.

Comments 7:  Section 4.2: Did the oxygen-impermeable chamber, in which the tissue samples were cultivated, contain some remaining air/oxygen in the beginning (i.e., was there some oxygen that may disturb hypoxic conditions at the start of the experiment)? Or did the authors remove gas/oxygen from the chamber?

Response 7: Thank you for this question. We do not remove the residual oxygen from the chamber. We recognize that there may be some residual oxygen in the chamber during the first few hours of culture. However, our experiments indicate that by 24 hours the cartilage would have reached a hypoxic state.

Reviewer 2 Report

Comments and Suggestions for Authors According to this paper, new, more representative, in vitro models for studing osteoarthritis are needed. In this paper authors propose a tissue-on-a-chip system based on patients tissue (three different patients). Authors showed different oxigen levels in the chip from bone to cartilage section (even if not quantified), cell viability up to 73 days and GAG, ALP, PIINP, Coll2-1 production over time. Authors collected explants from lesioned and non-lesioned tissues, but it seems that they analysed them all together. It would be worth to look at them also separately and to discuss if any differences emerged. It would be useful also to analyse tissue structure at the end of the perfusion time or at some end-points to evaluate if this Tissue-On-a-chip model allow analysis other than indirect evaluation of tissue stability/viability. According to the authors, no biomechanical stimulation is applied and this represent a strong limitation to be optimized in the future.

Minor issues: 4.1 Patients. Please report the number of protocol approval.

Figure 2 seems to have been stretched

Comments on the Quality of English Language

English need to be revised.

Author Response

Reviewer 2

Comments 1: According to this paper, new, more representative, in vitro models for studing osteoarthritis are needed. In this paper authors propose a tissue-on-a-chip system based on patients tissue (three different patients). Authors showed different oxigen levels in the chip from bone to cartilage section (even if not quantified), cell viability up to 73 days and GAG, ALP, PIINP, Coll2-1 production over time. Authors collected explants from lesioned and non-lesioned tissues, but it seems that they analysed them all together. It would be worth to look at them also separately and to discuss if any differences emerged. It would be useful also to analyse tissue structure at the end of the perfusion time or at some end-points to evaluate if this Tissue-On-a-chip model allow analysis other than indirect evaluation of tissue stability/viability. According to the authors, no biomechanical stimulation is applied and this represent a strong limitation to be optimized in the future.

 Response 1: We would like to thank the reviewer for their positive feedback and for all the useful comments and suggestions. When analyzing by groups the tissues with lesion and without lesion with respect to culture start, the significance is similar at the same days as when analyzing both groups together. On the other hand, comparing each group over time could be interesting; however, since this was not the aim of our research, we did not select all lesion grades (0-4) for each patient. Therefore, the results may not be conclusive.

Comments 2: Minor issues: 4.1 Patients. Please report the number of protocol approval.

Response 2: We apologize for the oversight. The number of protocol approval from the Institutional Ethics Committee has now been included in the revised manuscript (Section 4.1): “The study protocol was approved by the Institutional Ethics Committee (Comité Ético de Investigación Clínica, Hospital Clínico San Carlos, Madrid, Spain) in accordance with the principles of the Declaration of Helsinki, with study code 21/133-E.” (line 218-220) and in Institutional Review Board Statement (line 319-320).

Comment 3: Figure 2 seems to have been stretched.

Response 3: Thank you for pointing out the formatting issue with Figure 2. We have corrected the image distortion, and the revised figure has been updated in the manuscript